

# Genetic variability and structure of the Olive Field Mouse: a sigmodontine rodent in a biodiversity hotspot of southern Chile

Paulo S. Zepeda[1], Enrique Rodríguez-Serrano[2], Fernando Torres-Pérez[3], Juan L. Celis-Diez[4] and R Eduardo Palma[1]

[1] Laboratorio de Biología Evolutiva, Departamento de Ecología, Facultad de Ciencias Biológicas, Pontificia Universidad Católica de Chile, Santiago de Chile, Chile
[2] Departamento de Zoología, Facultad de Ciencias Naturales y Oceanográficas, Universidad de Concepción, Concepción, Chile
[3] Instituto de Biología, Facultad de Ciencias, Pontificia Universidad Católica de Valparaiso, Valparaiso, Chile
[4] Escuela de Agronomía, Pontificia Universidad Católica de Valparaíso, Valparaíso, Chile

Corresponding author
Paulo S. Zepeda, pszepeda@uc.cl

## ABSTRACT

The temperate rainforests of southern Chile, a recognized biodiversity hotspot, were significantly affected by Pleistocene glacial cycles in their southern portion and have been severely disrupted mainly due to recent human activities. Additionally, the landscape is characterized by a series of potential barriers to gene flow, such as the Chacao Channel, Cordillera de Piuche in Chiloé and both the Ancud and the Corcovado gulfs. We used mitochondrial DNA sequences and microsatellite data across several populations to evaluate the genetic variability and structure of the sigmodontine rodent *Abrothrix olivacea brachiotis*, one of the most common species of small mammals and an inhabitant of these biodiverse forests. Sequencing data showed that along with the recovery of high haplotype variation for this species, there was a low nucleotide diversity between haplotypes, showing no genetic differences between the Chiloé Island and continental populations in southern Chile or through any other geographic barrier in the study area. However, microsatellite data exhibited some level of population structuring. The most evident clusterings were those of the Chiloé Island and that of North Patagonia. These findings are corroborated by a barrier analysis that showed a genetic barrier in the latter areas, whereas the Chacao Channel was not a significant barrier for this rodent. Overall, the genetic variability and structure of *A. o. brachiotis* was concordant with historical factors, such as the Last Glacial Maximum and the presence of geographic elements that isolate populations.

## INTRODUCTION

The temperate rainforest of South America, mostly in Chile, is one of the largest areas of forests in the southern hemisphere and is rich in endemic species (*Veblen, 2007*). The region has also been named one of the 200 most important ecoregions by the World

Wildlife Fund (WWF) due to the biodiversity, endemism (particularly for plants) and degree of threat (*Olson & Dinerstein, 1998*). These forests have been included within the 34 hotspots of biodiversity in the world (*Mittermeier et al., 2004*). Approximately 45% of the pre-Columbian forest cover has been lost due to human activities (*Miranda et al., 2017*), being mainly replaced by crops, exotic forestry plantations (e.g., Pinus radiata and Eucalyptus spp), firewood extraction, overgrazing and recent fires (*Cavelier & Tecklin, 2005*; *Echeverría et al., 2007*; *Lara et al., 2012*; *Bowman et al., 2019*). Of note, the percentage of forest loss is stronger between 39° to 41°S in Chile, reaching its lowest value at 42°S in the northern portion of Chiloé Island. In addition, the type of vegetation replacement is different between the continent and the latter island, with more exotic tree plantations in the mainland and more shrublands in Chiloe.

Historical processes have modified the landscape, creating geographic barriers among populations and changes in population sizes. Among these barriers, there are the Piuche mountain range in Chiloé Island and the Gulf of Ancud and the Gulf of Corcovado that separates the island from North Patagonia. Alternatively, the Pleistocene glacial cycles affected an important area of the landscape in the Americas, particularly the southern cone of South America. Historical records show that the temperate rainforests of the Cordillera de la Costa in Chile played a central role in maintaining higher endemism, species and genetic diversity. Because the southern portion of South America was severely affected by the Pleistocene glacial cycles (*Holling & Schilling, 1981*; *Harrison, 2004*), the geographical ranges of several species associated with temperate rainforests were restricted due to climatic changes triggered by glaciations. The survival of most species was restricted to small areas of oceanic influence known as "refugia." In fact, several refuge areas have been recognized in coastal areas of south-central Chile, both for plants and animals (*Vilà et al., 2004*; *Sersic et al., 2011*; *Segovia, Pérez & Hinojosa, 2012*).

Within small mammals, the sigmodontine rodents (Cricetidae) are one of the most important inhabitants of the Chilean temperate rainforests, including *Abrothrix olivacea* (the "olive field mouse"), one of the most abundant and characteristic rodents of this biome. The species ranges from 18 to 54°S (*Mann, 1978*; *Osgood, 1943*; *Patterson, Teta & Smith, 2015*), encompassing areas of the Coastal Desert, the Mediterranean, the Valdivian and the Patagonian Forests and steppe. A former phylogeographic study reported strong structuring for the species, recognizing seven subspecies (*Mann, 1978*), including *Abrothrix olivacea brachiotis* restricted to the temperate rainforests of southern Chile, from Valdivia (39°S) southward to Aysén (45°S), including the Chiloé Island and southern archipelagoes nearby (*Osgood, 1943*; *Pearson & Smith, 1999*; *Rodríguez-Serrano, Cancino & Palma, 2006*; *Patterson, Teta & Smith, 2015*). The home range of *A. olivacea* in southern Chile is approximately 730 to 2,530 m$^2$ (*González, Murúa & Feito, 1982*), which is considered high for a small rodent (25 g average for adult forms). The subspecies *brachiotis* is found in diverse types of habitats from grasses to shrubs and humid forests (*Iriarte, 2008*). This subspecies is also found in Chiloé Island, regardless of the potential barrier that constitutes the Chacao Channel (widest portion 4.6 km and narrowest part 1.8 km). The results of nucleotide sequence analyses found no genetic differences between specimens from Chiloé (Northern part) and mainland areas (*Rodríguez-Serrano, Cancino & Palma, 2006*). The latter pattern

has also been reported for other sigmodontine mice, such as *Abrothrix manni* (*D'Elía et al., 2015*), the pygmy rice-rat or long-tailed mouse *Oligoryzomys longicaudatus* (*Palma et al., 2005*; *Palma et al., 2012*), and the microbiotheriid mouse opossum *Dromiciops gliroides* (*Himes, Gallardo & Kenagy, 2008*). The opposite pattern (i.e., differentiation between the continent and the island) has been reported for other species, such as the small cervid *Pudu puda* (*Fuentes-Hurtado et al., 2011*), Darwin's fox *Pseudalopex fulvipes* (*Vilà et al., 2004*; *Yahnke et al., 1996*), and even for an iguanid lizard, *Liolaemus pictus* (*Vidal et al., 2012b*). *Leopardus guigna* follows the same differentiation pattern of *Abrothrix olivacea*, at the subspecies level, when analyzed with mitochondrial sequences. However, further analyses with microsatellite data showed genetic structuring, recovering a Chiloe and a continental cluster, with the Chacao Channel creating a recent barrier to gene flow (approximately 8,000 years before present; (*Napolitano et al., 2014*). The latter scenario questions whether the use of a more sensitive molecular marker, such as microsatellites, can unveil a barrier condition for the Chacao Channel.

The information regarding genetic patterns of variability and structuring in the temperate rainforest of Chile has been mainly sourced by mitochondrial sequence analysis. The use of more variable markers such as microsatellites will provide valuable and sensitive data from a genetic scope (*Frankham, 1996*; *Frankham, Ballou & Briscoe, 2002*). With this type of marker, we can study patterns that may not be resolved with more conserved regions of DNA and will allow us to test for genetic variability and structure on a finer scale and from recent historical processes.

Using microsatellite and mitochondrial sequence data along several populations from both continental and island (Chiloé) areas of southern Chile, we evaluated the genetic variability and structuring of *Abrothrix olivacea brachiotis* along with its distribution in the temperate rainforests of southern Chile. In addition, we evaluated the potential effects of existing geographic barriers in the geographic distribution of *A. o. brachiotis*, such as the Chacao Channel, the Piuché Mountains in Chiloé Island, and the border of the LMG. The ecological and evolutionary features of *A. olivacea* populations, coupled with the hotspot of biodiversity where this species lives, provides a useful model for examining historical and ecological processes that shape the genetic variability and population structure of *A. o. brachiotis*.

## METHODS

### Sample collection

The study area encompasses the complete geographic distribution of the subspecies *Abrothrix olivacea brachiotis*, between Los Rios (40°S) and Aysen (45°S) regions in the temperate rainforest ecoregion (green shade with black stripes, Fig. 1). We sampled 11 localities with at least 2 sampling points, except for Chiloé National Park (Cucao), which had a single site (Fig. 1; see Table S1 for sampled localities). The sampling gap between the southernmost localities on the continent was due to the geographic difficulties in accessing that area for fieldwork.

Most of the tissues used in this study were retrieved from *A. o. brachiotis* voucher specimens deposited in the Colección de Flora y Fauna "Profesor Patricio Sánchez Reyes"

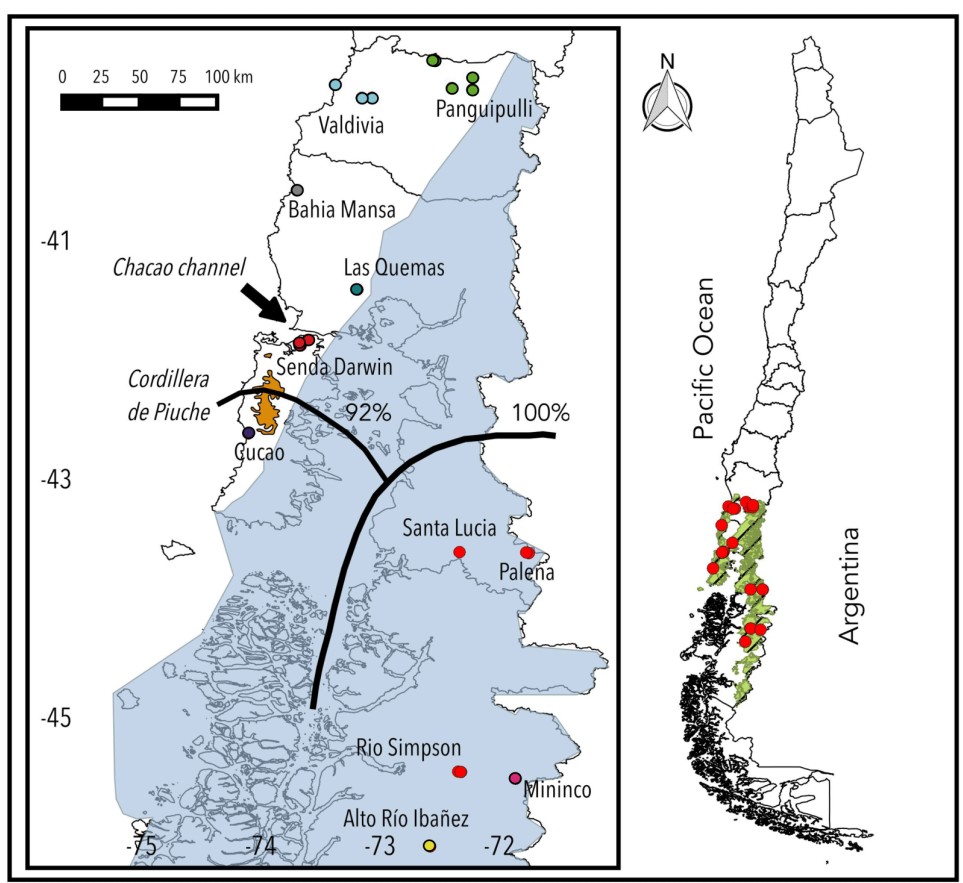

**Figure 1** **Map of sampling sites.** Study area in southern Chile. Blue shade represents the Last Glacial Maximum (LGM), green shade with black stripes in complete Chile indicates the distribution of the Temperate Rainforest. On Chiloe Island the Cordillera de Piuche is deliminated in orange. Black lines over the map indicates the result of the software Barriers, along with the percentage of bootstrap of each barrier. The colors on the sampling point of the localities (study area): Panguipulli (green), Valdivia (light blue), Bahia Mansa (gray), Las Quemas (Turquoise), Senda Darwin (dark red), Cucao (Purple), Mininco (pink) and Alto Rio Ibañez (yellow); represents localities that have mitochondrial sequences. Localities with light red color (Santa Lucia, Palena and Rio Simpson) were only amplified for microsatellite markers.

(SSUC), Departamento de Ecología, Pontificia Universidad Católica de Chile, Santiago, Chile, and in the Museum of Southwestern Biology (MSB), Department of Biology, University of New Mexico, Albuquerque, New Mexico, USA. The remainder of the samples were the result of field campaigns in which standard Sherman traps were used (8 × 9 × 23 cm; H. B. Sherman Traps, Inc., Tallahassee, FL, USA) with oats and vanilla as bait. Fieldwork protocols followed the standard bioethical and biosafety protocols outlined by the American Society of Mammologists (ASM; *Sikes, Gannon & The animal care and use committee of the American Society of Mammalogists, 2011*), and the Centers for Disease Control and Prevention (CDC; *Mills et al., 1995*), respectively. All of the new rodent captures were deposited at SSUC. The captures were conducted under the Chilean Government authorization (Servicio Agricola y Ganadero; SAG): Resolución Exenta

#5675/2013 and the approval of the Bioethics Committee of Pontificia Universidad Católica de Chile (#CBB-220/2012). A detailed list of specimens sequenced per locality is given in Table S1. A total of 187 samples were used in this study. All of them were genotyped for microsatellite loci, and a subset of 107 samples was sequenced for the mitochondrial hypervariable region I (HVI) analyses (see Table S1 for samples sequenced and genotyped).

## Laboratory procedures
### DNA Extraction

DNA was extracted from either frozen tissues (−80 °C) or ethanol preserved tissues (liver, lung or kidney) using the Wizard Genomic DNA Purification Kit (PROMEGA, Madison, Wisconsin) or the Phenol-Chloroform protocol (modified from *Sambrook, Fritsch & Maniatis (1989)*.

### Mitochondrial DNA

Through the polymerase chain reaction (PCR), we amplified the complete hypervariable domain I (HVI) of the mtDNA control region, using primers and protocols outlined in *Rodríguez-Serrano, Cancino & Palma (2006)*. A total of 483 base pairs (bp) of the HVI region from 92 individuals were sequenced and then deposited at the GenBank database (accession numbers: MH714355 to MH714446). In addition, another 15 sequences of the subspecies *Abrothrix olivacea brachiotis* were obtained from GenBank (Accession numbers: AY840064 to AY840078). We thus completed a total of 107 sequences that were aligned using the Clustal W program (*Larkin et al., 2007*) implemented in the software BIOEDIT (*Hall, 1999*). All samples were sequenced in MACROGEN Inc. (Seoul, Korea).

### Microsatellite DNA

A total of 28 microsatellite primer pairs were specially designed for this study through the company Genetic Marker Services Laboratories (London, UK). Following the manufacturer's instructions, we standardized only those that presented good quality in the report given by the company. A subset of 16 SSR was screened for polymorphisms and then tested for genotyping problems in Microchecker (*Van Oosterhout et al., 2004*). After these tests, 12 loci were the final set of primers used in this study (Table S2). Using the program FSTAT version 2.9.3.2 (*Goudet, 1995*), we tested for linkage disequilibrium between pairs of loci. PCR conditions were standardized specifically for these newly designed primers with a protocol in a 10 µl reaction volume containing 2 µl of Buffer 5X, 2 µl of MgCl2 at 25 µg/µl, 1 µl of dNTPs at 2 µg/µl, 0.25 µl of BSA, 0.2 µl of forward primer and 0.2 µl of reverse primer, 0.35 µl of M13 fluorescence tail, 0.15 µl of Taq polymerase, and finally 2 µl of DNA at 40 ng/µl. Each pair of primers was standardized to a thermocycler protocol with 32 cycles of 1 min denaturation at 95 °C, annealing of 1 min–2 cycles each 65–59 °C, 10 cycles at 58 °C, 10 cycles at 57 °C, an elongation of 1 min at 72 °C, and a final extension at 72 °C for 5 min (Table S2).

Microsatellites were fragment analyzed in an ABI3500 and then genotyped with the software GENEMAPPER (GeneMapper® v4.0; Applied Biosystems®, Foster City, CA, USA). In the standardization process, at least 8 samples of different localities were repeated to ensure consistency in the genotyping process. Additionally, the dataset was analyzed

**Table 1  Genetic diversity.**

| Population | Sampling locations | | | | | Genetic clusters | | | | | | |
|---|---|---|---|---|---|---|---|---|---|---|---|---|
| | N | Ho | He | A | Ar | N | A | Ar | Ho | Hs | Gis | Ne |
| Panguipulli | 14 | 0.835 | 0.887 | 12.33 | 10.45 | | | | | | | |
| Valdivia | 26 | 0.79 | 0.865 | 14.83 | 10.09 | | | | | | | |
| Bahia Mansa (BH_M) | 11 | 0.718 | 0.844 | 7.67 | 7.2 | 66 | 20.75 | 10.53 | 0.776 | 0.878 | 0.117 | 683.3 (313.9–∞) |
| La Picada | 4 | 0.562 | 0.844 | 4.33 | 4.33 | | | | | | | |
| Las Quemas | 11 | 0.775 | 0.84 | 9.75 | 8.57 | | | | | | | |
| Senda Darwin | 49 | 0.748 | 0.809 | 17 | 8.7 | 49 | 17 | 8.7 | 0.748 | 0.809 | 0.076 | 274.9 (164.1–762.1) |
| Cucao | 27 | 0.848 | 0.979 | 14.33 | 9.49 | 27 | 14.33 | 9.49 | 0.843 | 0.879 | 0.041 | 54 (41.1–76.4) |
| Palena Santa Lucia | 12 | 0.735 | 0.78 | 7.5 | 6.08 | | | | | | | |
| Rio Simpson | 9 | 0.749 | 0.831 | 8.08 | 8.08 | 45 | 15.66 | 9.13 | 0.759 | 0.825 | 0.08 | 113.2 (82.4–174.4) |
| Mininco | 9 | 0.762 | 0.829 | 7.67 | 7.5 | | | | | | | |
| Alto Rio Ibañez | 15 | 0.771 | 0.771 | 8.42 | 7.5 | | | | | | | |

**Notes.**

N, number of individuals; $H_o$, observed heterozygosity; $H_S$, expected heterozygosity; A, number of alleles; Ar, number of alleles with rarefaction (20 genes); $G_{IS}$, inbreeding coefficient; Ne, effective population size.
The two upper columns represent the grouping per sampling localities and per genetic clusters (according to STRUCTURE analysis).

with the software Microchecker (*Van Oosterhout et al., 2004*) for testing allelic dropout and null alleles.

## Analyses of genetic structure and variability
### Mitochondrial DNA

To examine population dynamics, we used Tajima's test of neutrality (*Tajima, 1989*). Assuming that sequence variation in the control region was neutral, we calculated the Tajima's D index to test the occurrence of population expansion if the values obtained indicated a significant negative value or a population equilibrium. To assess the population structure, a median-joining haplotype network was generated based on the HVI sequence dataset, using the software popART 1.7 (*Leigh & Bryant, 2015*; *Bandelt, Forster & Röhl, 1999*). The graph was calculated and plotted using the median-joining algorithm. Genetic diversity and variability were measured calculating haplotype (Hd) and nucleotide ($\pi$) diversity using popART 1.7 (*Leigh & Bryant, 2015*; *Bandelt, Forster & Röhl, 1999*) (Table 1).

### Microsatellite DNA

Genetic diversity indexes were calculated with the software GENODIVE (*Meirmans & Van Tienderen, 2004*) and HPrare (*Kalinowski, 2005*). We thus calculated the observed and expected heterozygosity, the number of alleles (A) and the number of alleles standardized to the population with the smallest sample size (Ar). Additionally, with the software NEstimator (*Do et al., 2014*), we estimated the effective population size (Ne) for each of the clusters calculated with the STRUCTURE software (see below) using the linkage disequilibrium method (*Hill, 1981*; Table 1). Deviations from the Hardy-Weinberg equilibrium were examined for each population at each locus calculating the fixation index Fis with the software GENODIVE. Genetic differentiation was assessed with pairwise Fst values between populations (Table S3). Isolation by Distance (IBD) was calculated
through a Mantel test with GenAIEx software (v6.5; *Peakall & Smouse, 2012*). To run this program, we generated a geographic distance matrix and a Nei's Index for obtaining a genetic distance matrix. Significance was tested using 9,999 permutations.

We used the STRUCTURE software (*Pritchard, Stephens & Donnelly, 2000*) to define population genetic structure without assuming a priori knowledge of sampling location or number of populations. To determinate the number of clusters, we used 10,000,000 iterations with a burn-in of 1,000,000 simulations. We evaluated K among 1 to 8 clusters, and each was run 10 times to ensure the stability and variance of likelihood values. Admixture and correlation models were used. These models assume mixed ancestry for all analyzed individuals and where the allele frequencies of closely related populations may be correlated. To determine the most likely K of our sample, we used the approach of Evanno implemented in the web platform Clumpak (*Kopelman et al., 2015*; *Evanno, Regnaut & Goudet, 2005*). In this platform, the second-order rate of change of the log probability of data with respect to the number of clusters was examined. The same platform delivers a graphic and color representation that combines runs.

We performed GENELAND analyses (*Guilliot, Mortier & Estoup, 2005*) considering the microsatellite genetic data plus the spatial location of samples. This approach is a Bayesian cluster analysis that uses individual geo-referenced genetic data to detect the number and geographic position of populations (*Guilliot, Santos & Estoup, 2008*). The algorithm identifies genetic discontinuities while estimating both the number and locations of populations without any a priori knowledge on the population units and limits. Once the number and limits of populations were established, the population membership probability was calculated from the posterior probability distribution of the MCMC. First, one independent run was performed by 10,000,000 iterations, sampling every 1,000 iterations of the Markov Chain and treating the number of populations as unknown varying between $K = 1$ to 8. Then, we chose the better of 20 independent runs, each of 10,000,000 generations and sampling every 1,000 but now treating the number of populations as a fixed parameter estimated from the first independent run. We used the admixture model and correlated allele frequencies to estimate the posterior probability of the data with 1 km of uncertainty to detect possible different clusters of samples with shared localities. From the posterior distribution, we drew a map of probability isoclines of population membership, one for each population or cluster inferred by the model. With this result, a map with the geographic limit was elaborated using QGIS (*QGIS Development Team, 2019*).

The software Barriers (vs 2.2 *Manni, Guérard & Heyer, 2004*) was used to test the occurrence of genetic barriers between different localities. This program uses a Monmonier's algorithm with Delaunay triangulation. The significance of a detected barrier is the result of 1,000 sub replicates, and the matrices of pairwise differences were generated by the software MSAT (*Dieringer & Schlötterer, 2003*). For this analysis, a barrier was considered valuable when we obtained bootstrap values over 90% (Fig. 1).

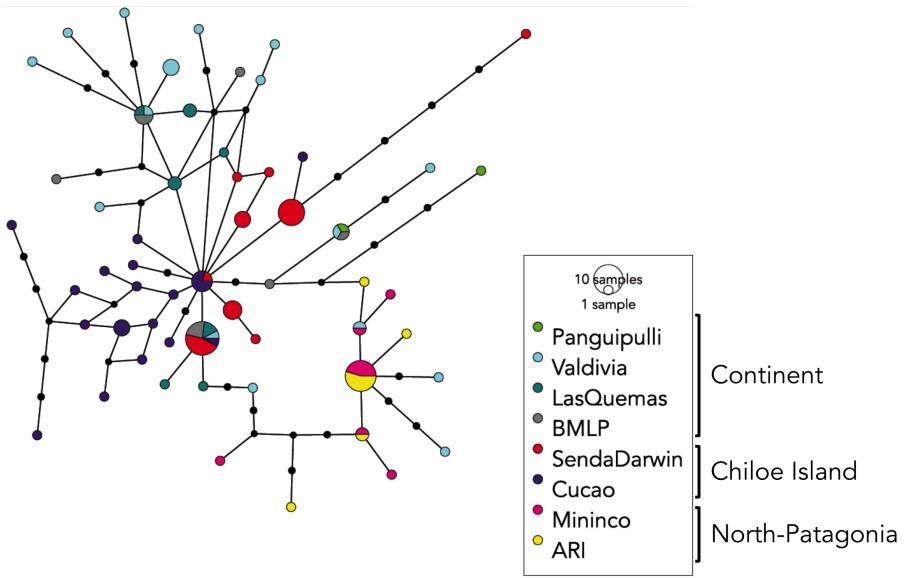

**Figure 2** **Haplotype network.** Haplotype relationships of *A. olivacea brachiotis*. Each circles represent a single haplotype sized in proportion of its frequency. The color code of each locality its shown in the figure (BMLP; Bahía Mansa and ARI; Alto Rio Ibañez).

## RESULTS

### Mitochondrial DNA data analysis

The haplotype gene diversity (Hd) was 0.965 (variance 0.00007), the nucleotide diversity (Pi) was 0.01070, and the Tajima index calculated over the mitochondrial data showed a value of $-1.84$ $(0.1\ p > 0.05)$, which is not significant, meaning that populations exhibited neutrality.

The median-joining network recovered 56 haplotypes (Fig. 2) for the 107 mtDNA sequences of the HVI mtDNA region. Figure 2 shows a group of haplotypes of low frequency and few mutational steps (between 1–3) from Cucao ("purple") in Chiloé Island. Another clustering of haplotypes ("red") was observed with samples from Senda Darwin Biological Station in the north of Chiloé Island. Some of these haplotypes exhibited high frequencies with up to 8 sequences but with low mutational steps (the highest with 5 steps). There is a shared haplotype of high frequency (13 sequences) that recovered localities from the mainland and the Chiloé Island. In addition, the Valdivian localities had haplotypes that spread throughout the network; these haplotypes were of low frequencies, with few mutational steps (1–3) and related to localities of Mininco and Río Blanco in the southernmost distributional range of *A. o. brachiotis* in northern Patagonia. At this latter point, a high frequency haplotype (11 sequences) was recovered that included the two latter localities of Mininco and Rio Blanco. Other haplotypes ("turquoise," "gray" and "green;" see legend Fig. 2) represent north-continental localities that are connected among them with few mutational steps and low frequencies.
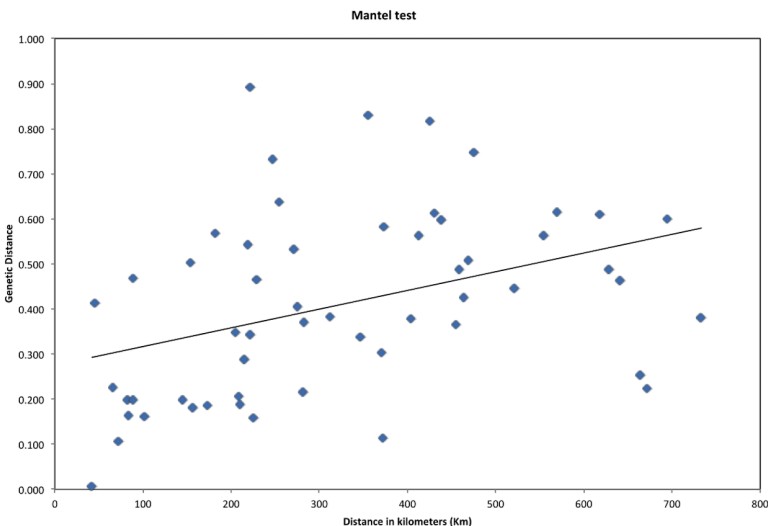

**Figure 3** **Mantel test.** Relationship between Nei's genetic distance and geographical distance in km among populations of *Abrothrix olivacea brachiotis* (Mantel test of correlation, rxy = 0.386, *p* = 0.01).

## Microsatellite data analysis

Results from Microchecker showed that loci Aol44, Aol55B, Aol91 and Aol52B deviated from Hardy-Weinberg equilibrium with heterozygosity deficiency. This was the case for only a single population per locus, and for this reason, data were not removed from the analysis. None of the populations revealed heterozygosity excess, and Microchecker did not identify genotyping errors, null alleles, or allele dropout in our data. There was no significant genotypic linkage disequilibrium detected for any pair of loci.

In Table 1, we show the genetic diversity for microsatellite data per localities. We considered 11 populations and pooled Palena with Santa Lucia due to the geographic proximity of the size of these two populations. Ho and He were similar except in La Picada, but this may be due to the low sample size. The FST index calculated between all populations gave a majority of significant differences between populations with the exception of La Picada vs. the rest, but this may also be due to a low sampling size. Additionally, the Mantel correlation coefficient, the test for isolation by distance, was positive (Rxy = 0.386, *p* = 0.01; Fig. 3), giving a positive and significant correlation between genetic and geographic distance.

We observed that the highest number of alleles was concentrated between the mainland and Senda Darwin Biological Station in Chiloé, although the Cucao and North-Patagonian populations also concentrated an important number of alleles. The effective number of alleles (corrected with the sample size) was higher in the continent than in Chiloé Island and North-Patagonian populations. The same Table 1 shows no significant differences between observed and expected heterozygosity values for all localities, showing high values (all of them near 0.8); Gis values were significant for all 4 localities.

Figure 4 shows the results of STRUCTURE analysis. The Evanno method (*Evanno, Regnaut & Goudet, 2005*) recovered five clusters (*K* = 5) with the highest probability

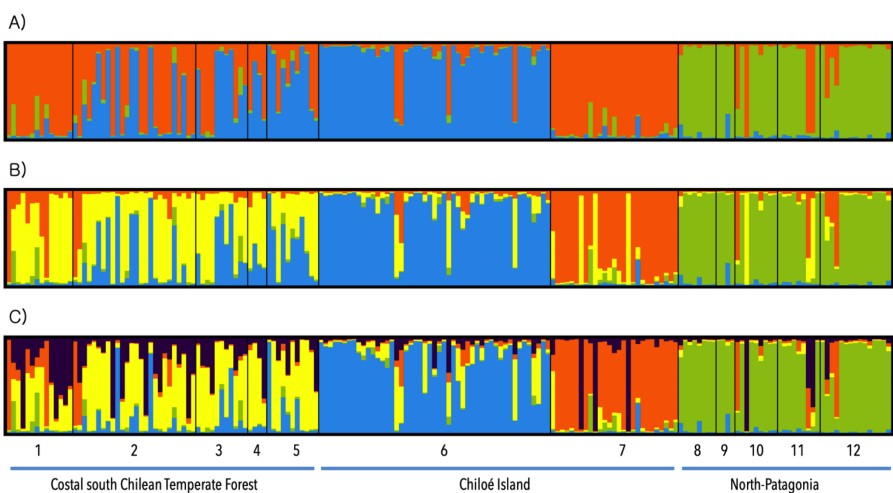

**Figure 4  STRUCTURE analyses.** Population structure of 187 individuals of *Abrothrix olivacea brachiotis* inferred from STRUCTURE analyses. (A) Structure with $K = 3$, second highest probability; (B) structure with $K = 4$; (C) structure with $K = 5$, highest probability with the Evanno method. Numbers represent the different sampling sites: 1, Panguipulli; 2, Valdivia; 3, Bahia Mansa; 4, La Picada; 5, Las Quemas; 6, Senda Darwin; 7, Cucao; 8, Santa Lucia; 9, Palena; 10, Rio Simpson; 11, Mininco; 12, Alto Rio Ibañez.

(Fig. 4C). From north to south, clusters 1 and 2 grouped populations from Panguipulli, Valdivia, Bahía Mansa, La Picada and Las Quemas ("purple" and "yellow"); cluster 3 ("blue") included mainly populations from Senda Darwin Biological Station (north of Chiloé Island); cluster 4 ("orange") constitutes the Cucao population (central Chiloé at Chiloé National Park), and the last cluster ("green") included southern populations of Palena, Santa Lucía, Río Simpson, Mininco and Alto Río Ibañez in North Patagonia. Clusters 1 and 2 ("yellow" and "purple") recovered an intermingling of probabilities for continental populations from the northernmost distribution of *A. o. brachiotis,* from Panguipulli to Las Quemas. However, some individuals from Panguipulli were assigned to the orange cluster. The second highest supported ΔK recovered 3 clusters (Fig. 4A), identifying the same clusters showed in $K = 5$: "blue," "orange" and "green," whereas individuals from the continent (clusters "yellow" and "purple") were assigned to clusters "orange" and "blue." We also show analysis with $K = 4$, exhibiting the same pattern, with coastal south temperate forest populations forming a not well defined cluster, two clusters in Chiloé Island, and another grouping the populations of North Patagonia.

Even though $K = 5$ clusters with the highest probability, $K = 4$ has greater biological sense. Following this finding, we estimated the effective size of $K = 4$ obtained with STRUCTURE (Continental, Senda Darwin, Cucao and North Patagonia). The biggest effective size was in the continent with 683.3, followed by Senda Darwin with 274.9; the smallest cluster was Cucao with 54 individuals. In the mainland, the top limit of the confidence interval is infinite (Table 1).

The population structure for *A. olivacea brachiotis* was also evaluated using the GENELAND v. 1.0.7 program. This analysis inferred three genetic clusters in the study area. A first cluster (Fig. 5A) mostly grouped individuals representing localities of continental

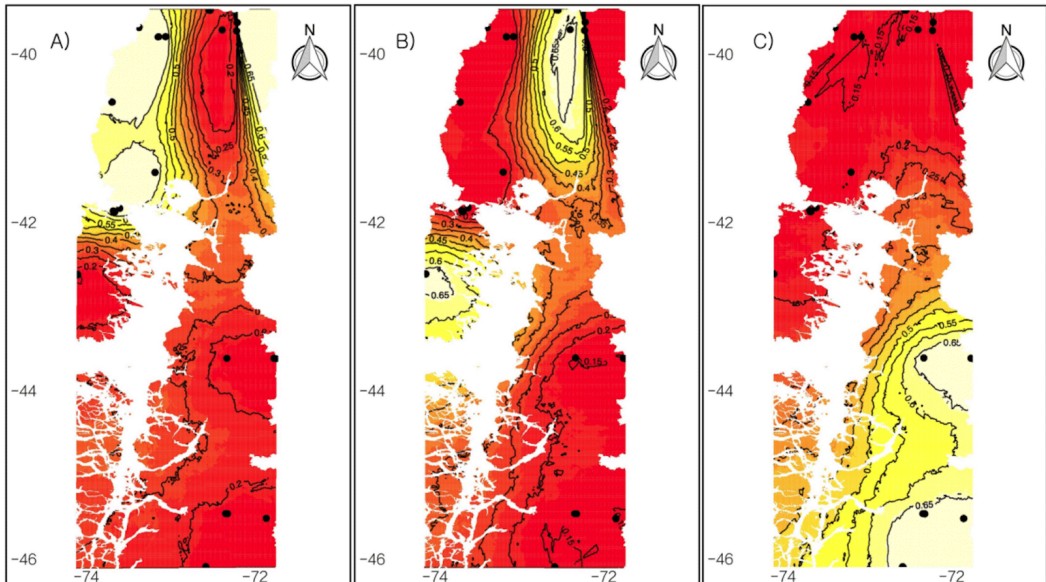

**Figure 5 Spatial population structure.** GENELAND analyses with posterior probability isoclines. Black dots represent the locality presents in this study. Yellow indicates regions with he greatest posterior probability of inclusion, and red areas shows region with lower probabilities of inclusion, this proportional to the degree of coloring. (A) Cluster number 1 with populations of Valdivia, Bahia Mansa, Las Quemas, La Picada, Panguipulli and Senda Darwin (Chiloe); (B) Cluster number 2 with populations of Panguipulli and Cucao; (C) Cluster number 3 with populations of Santa Lucia, Palena, Rio Simpson, Mininco and Alto Rio Ibañez.

Chile and the northern portion of the Chiloé Island. A second cluster joined individuals representing localities of Panguipulli and Cucao (Fig. 5B), and a third cluster grouped all southernmost populations (Fig. 5C). All clusters presented 0.65 of probability in the isocline that grouped the populations.

The program Barriers corroborated the genetic isolation of clusters by barriers. In fact, a strong barrier is hypothesized between populations of North Patagonia and the rest of *brachiotis*' distribution with 100% of appearance with 1,000 replicas, and a second barrier was proposed in Chiloé Island between Senda Darwin Biological Station and Cucao. Both proposed barriers exhibited high levels of consistency when we performed the analysis with 1,000 replicates.

## DISCUSSION

Using both mitochondrial and microsatellite markers, we evaluated the structuring and variability across the distribution of *Abrothrix olivacea* in the temperate rainforests of Southern Chile. Our results corroborated the phylogeographic study of *Rodríguez-Serrano, Cancino & Palma (2006)*, which showed no difference when comparing populations of *A. o. brachiotis* between Chiloé Island and the mainland using mitochondrial sequences. In the present study, we recovered two new haplogroups, one in the south of Chiloé (Cucao) that shares haplotypes with the continent, and the North Patagonia haplogroup that included

populations of Mininco and Alto Rio Ibañez (ARI). The last haplogroup had one shared haplotype between Valdivia (Northern continental distribution) and Mininco, and few mutational steps with other Valdivian haplotypes. In summary, the mitochondrial analysis showed low genetic structuring for the subspecies *brachiotis* and the nonoccurrence of exclusive haplotypes for any of the localities and recovered shared haplotypes with North continental localities. The overall mtDNA network had low nucleotide diversity, meaning that besides the high number of haplotypes (high haplotype diversity), there were little differences between them.

The Microsatellite Genetic diversity indices showed that across all populations sampled (11), there were no significant differences and Ho versus He was similar with the exception of La Picada, although this may be due to small sample size. The pairwise Fst showed that only close localities did not show significant differences, such as Mininco with Rio Simpson in North Patagonia. This finding is consistent with the pattern of isolation by distance found in the data. Additionally, the La Picada population, when compared with other localities, did not show (in general) significant differences, most likely due to small sample size.

In general, the number of alleles standardized for the sample size revealed that across populations, they all have similar values of allelic richness (7.20–10.45). The only population that exhibited a lower number of alleles was La Picada (4.33). When populations were grouped into four clusters (Continental, Senda Darwin, Cucao and North Patagonia), the difference was even smaller (8.70–10.53). Senda Darwin in Chiloé had the lowest allelic richness and the continental groups had the highest. This fact may be due to the recent separation of this population (Senda Darwin) from the continent. High continental allelic richness may explain the different sources of alleles: more than one refuge represents the genetic diversity of this cluster. In fact, this cluster also exhibits the highest number of private alleles. In the case of *A. o. brachiotis*, we did not find a heterozygosity deficit in any of the localities, as there were no significant differences when comparing mainland and island populations. Thus, it seems that despite recent intense forest fragmentation and degradation (*Echeverría et al., 2007*), the habitat for *A. o. brachiotis* does not seem to be a critical factor for the genetic variability, as it can adapt to new vegetational types to preserve heterozygosity (*Murúa & González, 1982*; *Murúa & González, 1986*; *Patterson, Meserve & Lang, 1990*; *González, Murúa & Jofré, 2000*).

Microsatellite data analyzed across *A. o. brachiotis*'s geographic range recovered five genetic clusters with STRUCTURE, with those results having the highest probability through the Evanno method. Two of these clusters corresponded to populations from the mainland, two others corresponded to populations from Chiloé Island, and a single cluster was restricted to the southernmost limit of the subspecies range in the North Patagonian forests (Aysen region). The clusters recovered in the continent ("yellow" and "purple;" Fig. 4C) were not well defined, and individuals were not assigned to a single cluster. They had mixed probabilities belonging to one or another cluster ("yellow" or "purple"), suggesting that these populations maintained their genetic variation and served as a source of alleles for other clusters (*Omote et al., 2012*). In an LGM scenario, particularly during the interglacial period, coastal refuges and dispersalist populations of iced eastern localities

may have dispersed back to central depression and pre-Andean areas in southern Chile (e.g., Panguipulli area). This may have allowed the intermingling of coastal and eastern populations as suggested by the pattern of the current STRUCTURE analysis (*Sersic et al., 2011*). The pattern of structure for the mainland agreed with the high allelic richness and number of private alleles, due to the composition of genetic diversity of the area, on which more than one refugia has been recognized, which might suggest the source of variation.

The structuring of *A. o. brachiotis* within Chiloé is more evident. We recognized two genetic clusters in the island, one represented by the northern population of Senda Darwin Biological Station, and the other represented in the west-central portion of the island, the Chiloé National Park at Cucao. These genetic structures of *A. o. brachiotis* in Chiloé might be explained by different but not mutually exclusive causes, such as the Cordillera de Piuché (a portion of the Coastal Cordillera that separates the Chiloé National Park from the northernmost portion of the island where the Senda Darwin Biological Station is located), and the major town of Castro (Chiloé's capital city) towards the east of the Coastal Cordillera. Other additional barriers that might have isolated the Cucao area from the rest of the island are historical events, such as the glacial border of the LGM during the Pleistocene that advanced throughout the island. This glacial border left a refuge area in northwest Chiloé where a portion of the National Park is located (*Rabassa & Clapperton, 1990*; *Moreno et al., 2015*; Fig. 1). These current and historical factors might have contributed to the existing genetic structure of *A. o. brachiotis* populations within the island. To the southeast of Chiloé, in the mainland, the North Patagonian structuring of *Abrothrix* in Palena and Aysén may be explained by isolation by distance: their geographic isolation with respect to Chiloé, and the geographic distance with respect to mainland populations located to the north. These facts, coupled to the glaciation of the LGM that completely covered these populations, might have forced *A. o. brachiotis* to find refuge areas separated from other populations.

The effective size of the different clusters is consistent with the size of the localities. We have the largest population in the mainland cluster that has a wide range of sampling points. Moreover, Cucao is estimated as the smallest Ne and is the population that shows the sharpest isolation, with the Piuche Mountains on one side, and the condition of glacial refugia. Of note, the analysis estimated that the continental cluster has the highest effective size. However, its upper limit as infinite could be due to a large effective size, or a limitation of our dataset, considering that this cluster is the largest, and it may need a more exhaustive sampling.

Geneland analyses, on the other hand, recovered 3 clusters. One cluster is shared by populations of the continent and Senda Darwin Biological Station in Chiloé Island. This cluster sustains the hypothesis that the Chacao Channel is a recent barrier for *Abrothrix* since the sea level descended during LGM, allowing for a glacial bridge between Chiloé and the continent, generating a refuge area in the coast of the Chile Lake district (*Sersic et al., 2011*). Genetic connectivity between Chiloé and the continent have been inferred for other mammals, such as the microbiotheriid mouse opossum *Dromiciops gliroides* (*Himes, Gallardo & Kenagy, 2008*), the long-tailed sigmodontine mouse *Oligoryzomys longicaudatus* (*Palma et al., 2012*), the sigmodontine *Abrothrix manni* (*D'Elía et al., 2015*),

and even the iguanid lizard *Liolaemus pictus* (*Vidal, Moreno & Poulin, 2012a*; *Vidal et al., 2012b*). Geneland also recovered two different cluster populations in the island, one to the north of Chiloé (Senda Darwin) and the other in central west Chiloé (Cucao), consistent with the results of STRUCTURE. The latter agreed with the hypothesis of refuge for Cucao and the effective barrier that constitutes the Piuché Mountains. Finally, the third cluster is the same that STRUCTURE analysis recovered for North Patagonia populations.

Finally, the Barrier analysis (*Manni, Guérard & Heyer, 2004*) corroborated the occurrence of a genetic barrier due to the Cordillera de Piuché in Chiloe. The same analysis hypothesized the existence of a barrier between North Patagonia populations and the rest of *brachiotis*' distribution. Both barriers might also be inferred with STRUCTURE analysis (a distinctive cluster for northern Chiloé, Cucao and North Patagonia) and with Geneland, which show clusters that are separated by the aforementioned barriers. The Barrier analysis also demonstrated that the Chacao Channel would not be a genetic barrier for *A. o. brachiotis*, and these results are consistent with those of Geneland (Fig. 5A)

## CONCLUSIONS

In this study, we obtained a subtle structuring of populations at the northern portion of the continental range of *A. o. brachiotis*, which is more evident in Chiloé with two major clusters within the island. An additional structuring was recovered in the northern portion of the Patagonian range of *A. o. brachiotis*. Our results seem to suggest that *A. olivacea* and particularly *A. o. brachiotis* restricted to the temperate rainforests of southern Chile, reacts more to historic than to ecological events regarding its genetic population dynamics and that interruptions of connectivity (Chacao Channel) would not constitute a genetic barrier for this rodent. The genetic structuring observed would respond to the geographic and long dated isolation of populations, as observed by the presence of Cordillera de Piuché that isolated populations in north Chiloé (Senda Darwin) with respect to those of Parque Nacional Chiloé (Cucao). In addition, populations from the latter geographic area are located in what was a glacial refuge during the Pleistocene. North Patagonian populations, on the other hand, may constitute a genetic clustering due to their isolation with respect to Chiloé populations and the geographic distance from the northern range of this subspecies. This isolation might be enhanced by geographic barriers, such as both the gulfs of Ancud and Corcovado and the mountain range of the Andes.

## ACKNOWLEDGEMENTS

We acknowledge the laboratory support of R. A. Cancino and M. P. de Castro from the Laboratorio de Biología Evolutiva (Departamento de Ecología, P. Universidad Católica de Chile), and G. Peralta from the Laboratorio de Diversidad Molecular (CONYCIT—FONDEQUIP EQM150077; same Department) for her support with the genotyping analyses. This paper contributes to the research program of the Senda Darwin Biological Station (Chilean LTSER Network).

 

### Funding
This work was supported by the Fondo Nacional de Ciencias y Tecnología (FONDECYT, Chile) through the following grant numbers 1130467, 1171280, 1170486 and 1170761. The funders had no role in study design, data collection and analysis, decision to publish, or preparation of the manuscript.

### Grant Disclosures
The following grant information was disclosed by the authors:
Nacional de Ciencias y Tecnología (FONDECYT, Chile): 1130467, 1171280, 1170486, 1170761.

### Competing Interests
The authors declare there are no competing interest.

### Author Contributions
- Paulo S. Zepeda conceived and designed the experiments, performed the experiments, analyzed the data, prepared figures and/or tables, authored or reviewed drafts of the paper, approved the final draft.
- Enrique Rodríguez-Serrano and Fernando Torres-Pérez contributed reagents/materials/analysis tools, authored or reviewed drafts of the paper, approved the final draft, funding.
- Juan L. Celis-Diez authored or reviewed drafts of the paper, approved the final draft, funding.
- R Eduardo Palma conceived and designed the experiments, contributed reagents/materials/analysis tools, authored or reviewed drafts of the paper, approved the final draft, funding.

### Animal Ethics
The following information was supplied relating to ethical approvals (i.e., approving body and any reference numbers):

Pontificia Universidad Católica de Chile provided full approval (code CBB-220/2012) for this research related to animal care and bioethical protocols.

### Field Study Permissions
The following information was supplied relating to field study approvals (i.e., approving body and any reference numbers):

Servicio Agricola y Ganadero (SAG, Chile) provided a capture permit (#5675/2013) for the samples used in this study.

### Data Availability
The sequences used are available in Genbank: accession numbers MH714355 to MH714446.

## Supplemental Information

Supplemental information for this article can be found online at http://dx.doi.org/10.7717/peerj.6955#supplemental-information.

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
