# Peer review of "Genetic variability and structure of the Olive Field Mouse: a sigmodontine rodent in a biodiversity hotspot of southern Chile"

_PeerJ, doi:10.7717/peerj.6955_

## Round 0.1 · original submission · Major Revisions

I have now received three excellent and very detailed reviews for your paper. All reviewers agree that your paper is generally well-written, interesting and important and that your dataset is good. However all of them highlight that you do not explicitly test the effect of habitat change nor historical factors on genetic variability or population structure. I would therefore request you either change your initial aims and focus exclusively on population structure, or try to quantitatively relate genetic diversity and population structure to habitat loss and historical factors. In the later case, please consider performing the additional analyses suggested by Reviewer 2 (IBD and Barrier tests).

When revising you paper, please carefully address each one of the raised issues. Please modify the abstract and introduction to include specific, testable hypotheses with clear predictions, and make sure all tests and protocols are mentioned in the methods. In the discussion, avoid conclusions that are not strictly supported by your data, and please address the effect of possible hybridization with adjacent but unsampled subspecies.

·

Basic reporting

This is a generally well-written piece. I am frustrated, after editing the entire ms. for English fluency, that my Abrobat failed and no text corrections could be saved. It needs minor attention throughout

Literature was generally present although two references to my own work were missed and would improve the authors inferences: one regarding the current taxonomy and distribution of Abrothrix olivacea (the authors cite Osgood 1943 and Mann F 1978, which both consider xanthorhinus a distinct species and brachiotis as the southernmost subspecies; in reality, hybridization across the brachiotis-canescens boundary is important and may account for the discovery of geographic structure). The current taxonomy of this SPECIES (i.e., Abrothrix olivacea, not A. o. brachiotis, which is a subspecies!) is given here--Patterson, B. D., P. Teta, and M F Smith. 2015 Genus Abrothrix Waterhouse, 1837. In Mammals of South America, Vol. 2: Rodents. J.L. Patton, U.F.J. Pardiñas, and G. D'Elía, eds. Pp. 109-126. Chicago: University of Chicago Press.

A critical part of the analysis was left unsaid: how is Abrothrix olicavea brachiotis distinguished? Species have reproductive and historic limits, but subspecies by definition intergrade. What is the effect of intergradation, or hybridization, at its boundaries with other Abrothrix olivacea subspecies affecting the patterns reported in this paper? This is not discussed anywhere in the paper.

Another major problem with the article as written is that Abrothrix olivacea would be among THE WORST species to study if anthropogenic deforestation was really a primary target of the article. Its presence in a host of habitats (they later cite Murua and Gonzalez 1982 but see also Patterson et al. 1990 Quantitative habitat associations of small mammals along an elevational transect in temperate rainforests of Chile. Journal of Mammalogy 71(4):620-633.) and throughout the offshore archipelagos (Guaitecas and Chonos) is testament to its superb colonizing ability and habitat tolerance. Only Oligoryzomys longicaudatus is more ubiquitous!! Why not just focus on understanding the population structure of this forest-associated species, without all the conservation hoopla of hotspots and deforestation, which is distracting and potentially even misleading?!

Experimental design

Appears strong. One could wish for longer sequences of a gene-encoding locus, but HV1 appears suitable for the present analysis.

Validity of the findings

Generally appear valid, but the authors need to reconcile their results, especially with respect to geographic subdivisions of the range, with the potential for hybridization with adjacent but unsampled subspecies of A. olivacea (see Fig 4 and the differentiation of the Aysen populations where A. o. brachiotis and A. o. canescens come together). The Geneland analyses apparently build topography into their predictions, but the extrapolations of distributions into Argentinian territory to the east of the Andes looks suspect

line 53: "Abrothrix olivacea brachiotis, one of the most common species of small mammals"--it is a subspecies, not a species
line 78: "second largest area of temperate forests in the world"--temperate rainforest (the taiga of Canada and Russia is far larger)
line 85: "overgrazing" -- isn't a far greater threat to native forest species clear-cutting for pasture?
line 106: citations: Both these authors recognized Abrothrix xanthorhinus in the south, and so their treatments of A. olivacea do not extend S of 45 degrees. Better to cite here
Patterson, B. D., P. Teta, and M F Smith. 2015 Genus Abrothrix Waterhouse, 1837. In Mammals of South America, Vol. 2: Rodents. J.L. Patton, U.F.J. Pardiñas, and G. D'Elía, eds. Pp. 109-126. Chicago: University of Chicago Press.
which give range extensions and modern taxonomy
line 116: specify locus used? This statement of variability means different things for HV1 and Rag1?
line 119: Himes et al 2008. see also
Monotypic status of the South American relictual marsupial Dromiciops gliroides (Microbiotheria) Elkin Y Suárez-Villota; Camila A Quercia; José J Nuñez; Milton H Gallardo; Christopher M Himes; G J Kenagy. Journal of Mammalogy, Volume 99, Issue 4, 13 August 2018, Pages 803–812, https://doi.org/10.1093/jmammal/gyy073
line 121: Although there has not been genetic confirmation of no differentiation, there is also the distribution of Loxodontomys micropus fumipes, known from Chiloé and Santa Cruz Patagonia
line 124: another Chiloé endemic is the nutria: Myocastor coypus melanops,
https://en.wikipedia.org/wiki/Coypu
line 140: The most critical question here is "how were individuals of "A. olivacea brachiotis" distinguished form the other 6 subspecies of A. olivacea recognized by Mann, Osgood, etc?" In other words, how do we know that the patterns of genetic variation you are describing do not result from hybridization with other adjacent subspecies, which have different habitat distributions and different population histories?
Line 215: choose-->chose
Line 311: isn't the geographic replacement of Abrothrix manni (northern Chiloé) and Abrothrix sanborni (southern Chiloé) worth mention here, to corroborate the compound history of the island?!
lines 324-326: the authors make this look like a conclusion from their analysis, whereas an even-handed examination of its wide distribution and generalized habitat in the Introduction would have made it expected
line 341: Anthropic effects=anthropogenic effects
line 347: role of hybridization?
line 357: A. sanborni relevant here
line 368: sea level lowering
line 380: the negligible effect of ...
line 383: this is where Patterson, Bruce D., Peter L. Meserve, and Brian K. Lang.
1990 Quantitative habitat associations of small mammals along an elevational transect in temperate rainforests of Chile. Journal of Mammalogy 71(4):620-633. could be cited
line 394: heterozygosity

Additional comments

This is a useful analysis that should be published. There is no need to argue beyond the analysis (e.g., conservation importance of its results). I would instead argue that A. o. brachiotis is co-distributed with a majority of species inhabiting the south temperate rain forests and that factors affecting its population structure may be general and applicable to many of them.

Reviewer 2 ·

Basic reporting

Some sentences need to revise to improve the flow. I have made some suggestions in general comments.

Experimental design

The authors of this work investigate through mtDNA sequences and 12 microsatellite loci the genetic diversity and structure across 11 sampling sites of a sigmodontine rodent in the temperate forest in Chile. The assumption is that historical and anthropogenic factors have shaped the genetics of this species, which is found in continental and island locations. The authors state that historical factors rather than ecological have shaped the genetic patterns observed in this study. The study is within an interesting and important biodiversity hotspot in South America, as such it contributes with valuable genetic information for an area where more studies are needed. However, I found the introduction and the abstract (last sentence) misleading. The introduction mentions important likely effects of habitat fragmentation on genetic diversity and structure due to human activities in this region, however, there is no explicit test to address the effects of landscape change such as habitat fragmentation. Also, there are assumptions on geographic barriers (island vs mainland; the Cordillera de Piuchén) and geographic isolation on genetic patterns, but no explicit test for those were carried, such as an IBD or Barrier test. Thus, the abstract and introduction should be restructured to make clear the objectives that were explicitly tested.
Second, methods also need restructuring, some test were not mentioned in methods, but mentioned in results, or information is missing such as details on sampling, laboratory procedures and data analyses.

Validity of the findings

The discussion also needs to be changed as the authors conclude on factors that were not tested, such as anthropogenic effects, or geographical isolation (this last one, which can be easily addressed with IBD test). The conclusion thus lack data support based on the analyses carried.

Additional comments

Ln48-51: This sentence is long and not much clear, divide the ideas.

Ln52; loci to evaluate the genetic variability across several populations of the sigmodontine rodent

Ln53: Abrothrix olivacea brachiotis, a common small mammal species of these disrupted forests

Ln54-56: Here again separate ideas in two shorter sentences.

Ln58: latter island? I don’t understand this.

Introduction

Ln63: the word disruption can mean disturbance, interruption, modification, etc. After reading the first paragraph I think the authors ideas are more specific to habitat fragmentation, which affects connectivity. So, I suggest to better change the term to habitat fragmentation since this is widely used in the scientific literature

Ln64: of the genetic diversity. I would say genetic diversity of populations rather than a species
Ln65: Since the sentence refers to loss of genetic diversity due to habitat fragmentation, I suggest to better include reference that directly assessed these effects such as meta-analyses carried in animals and plant taxa (see Aguilar et al. 2008 Mol Ecol Dec;17(24):5177-88)

Ln65: Loss of genetic variation

Ln69: I would say better environmental change

Ln88: So, readers know Chiloe is an island: the north population of the Chiloe island is located.

Ln93: Among these, the Pleistocene glacial cycles affected an important area of the landscape in the southern portion of South America

Ln96: higher than? Otherwise just use high

Ln104: important species

Ln107: remove former

Ln116: Be more specific of the type of DNA marker used (mtDNA sequences)

Ln125-127: Why is this interesting?

Ln127: however, microsatellite data showed genetic differentiation between the Continent and the Chiloe island, which implies that the Chacao channel is a recent barrier to gene flow (about 8,000 years BP) (Napolitano et al. 2014).

Ln130-136: Be more specific about the aims of the manuscript. State in which way this work is different from that of Rodríguez-Serrano et al. 2006 not only in the type of molecular markers used but, on the question, asked (e.g. contemporary patterns of gene flow). Also, a large portion of the introduction states the effects of habitat fragmentation due to human activities (including the abstract), but the Chacao channel as a barrier to gene flow (when the comparison is made with the Leopardus guigna study) is not a human-made factor, but a geographic factor. From the information presented I can see this is a fragmented landscape, both natural and human-made, but the methods do not explicitly test any anthropogenic landscape effect on genetic diversity or structure, such as habitat fragmentation, agriculture, or urbanization. The introduction should be restructured to make clear this is ms is not about investigating anthropogenic effects on genetic structure.

Methods

Sampling. Explain better the distribution of the subspecies and if the sampling covers mostly its distribution or if gaps exist and where. Why for instance, are some sampling sites very nearby (Senda Darwin, Valdivia, Panguipulli), and others separated by more than 50km (Alto Rio, Mininco)? I understand the limitations of an extensive sampling, but information on this should be provided so the reader has an idea of the scope of the results.

Year of sampling collection?
Why if 187 samples were collected, 92 individuals were DNA sequenced? And how many samples for microsatellites were processed?

For DNA sequences and microsatellites how many samples were repeated to calculate sequencing and genotyping error rate, in particular for microsatellites?
Since the 12 micros are new, it is important and valuable to other readers to show the table not within appendix but within the main text with information such as bp size, motif, and genetic diversity parameters.

Does the alleles for the micros were observed through fragment analyses (platform)?
Which software was used for microsatellite genotyping?

The data analyses would be clearer if they are split by molecular marker, there are some methods specifically for each marker type and some for both.

Ln193: which parameters were used to calculate Tajima´s D. If a population is in equilibrium which value is expected? If positive?

Ln199: usually the standard is at least 10 runs per K

Ln201-202: This is a popular method, so this info can be omitted (these models assume..)

Ln204: How the output (barplots) were observed and created?

Ln209: genetic populations

Ln219: Again, four replicate runs may be few. And how the output maps were created?

Ln225: This paragraph more than statistical analyses is on genetic diversity parameters and it goes first than any other method.

For the microsatellites, since they are new, other important test should be conducted such as Linkage disequilibrium. Are some loci that show LD more frequently? If so, then some loci should be discarded if STRUCTURE analyses are done. Otherwise, if LD and HWE are present, other multivariate methods such as PCA or DAPC can be performed which are free of LD and HWE assumptions.

Also, is there evidence of null alleles, allele drop out or other genotyping bias such as stutter peaks? if so corrections should be done for genetic diversity and differentiation parameters.

Ln231-232: How pairwise FSTs were calculated (software)? And only for microsatellites? Both pairwise FSTs for mtDNA and micros across the 12 localities can be estimated and compared.

Since there are some regions that show genetic structure with microsatellites and that later these are discussed to be related to some geographical features, it can be performed other test such as AMOVA or SAMOVA, to test how much genetic variation can be explained by the different geographic regions. Other test that specifically test for barriers are BARRIER, which uses a Monmonier algorithm to spatially detect barriers across a geographic area. This can be a test to provide support to the assertions on the discussion, which were not directly tested.

On the other hand, is there evidence of isolation by geographic distance?


Results.

Would be more informative to show genetic diversity parameters for both markers within a table per sampling locality.

Ln240: that populations show neutrality? But the Tajima D was calculated for the total of populations and not for each population or K genetic cluster, which may have a different history?

Ln246: Is not very clear the description of this result. Usually will start stating which is the central haplotype(s) that has the highest frequency and from which regions are. When it is stated a haplotype has low mutational steps, respect to which another haplotype? Referring to low or high mutational steps is always in a pairwise comparison, otherwise is not clear what this means.

Ln259-265: Although the most probable number of K is 5 according to the Evanno method, from the barplot it can be seen is 3 or 4, what about barplot k=4? There is high admixture across locations 1 to 6.

Ln278-295: move this to the beginning of the results. Also, all methods reported in 290-295 were not mentioned in methods. Every method should be stated in methods first. These results are the first one to be reported in the section.

Table 1 and 2: Why not to calculate pairwise FST and genetic diversity across the 12 localities. The FST values obtained would support either the results from structure or not.

Fig. 1 The blue shade can be confusing at first glance as it gives the impression that the blue is actual land. I think the best is to send it back to the map or with transparency so the lines with the limits of the continent and the archipelago looks more marked. However, I don´t have much clear how this LGM representation is specifically used in the ms to make a point to discuss.

Also in the larger map of whole Chile, spot the distribution of the temperate forest (Maule and Aysen) so the reader can see this data on the map.

Maybe fig 1 and 2 can be combined, the dots if sampling localities with the same color of the haplotype network to better have a visualization of the results of fig. 2.

Fig 3. Which are the probability of membership of these 3 clusters? If not provided is not possible to judge if they are likely true clusters

Discussion

Ln316-326: there is no real evidence on this, as it was not explicitly tested: I don’t agree it is safe to state there is no effect of habitat loss as either the sampling nor the analyses performed allow to conclude this.

Ln327-336: I don’t agree there is five genetic clusters, that’s what one analyses showed, but the membership probability states other thing. These ideas are redundant with results, I think this paragraph explain the result better that the one presented in results.

Ln335-339; But what about the Tajima D result? It does not indicate population expansion.

Ln340-345: There is no real evidence for this statement. Besides, there is no data collected on the size of populations, neither on the migration routes. Contrary, I think there is evidence of gene flow as it is seen on Structure and GENELAND from locations 1 to 7.

Ln349-360. But also, which is the explanation locations Panguipulli and Cucao are highly genetically related to be in one cluster (there is no explanation to this)? And that instead there is two clear genetic clusters within the island?

Ln363-365: But not isolation by distance was tested, and it is a very easy test to do.

Ln367-368: I don’t understand how this cluster sustain the hypothesis the channel would be a barrier, if these localities are within one single cluster? Is not the opposite, not a barrier?

Ln380-382. Same to previous comment, no factual evidence.

Ln399: But no IBD test performed.

Reviewer 3 ·

Basic reporting

This article should be revised to better communicate hypotheses, results, and interpretation of results. Some of the language used was overly vague. More suggestions related to Basic Reporting are included in my General Comments to Authors, below.

Experimental design

Yes, the project appears to have been designed reasonably well. Additional analyses would be helpful, however, including the inclusion of a phylogeny. More suggestions related to experimental are included in my General Comments to Authors, below.

Validity of the findings

All of the analyses appear to have been done correctly. Some of the conclusions are obvious and well supported by the data. Some conclusions regarding the impact of humans is especially speculative and not tied to a specific hypothesis. More details are provided below.

Additional comments

In this study, the authors investigate the population structure within a portion of the range of Abrothrix olivacea in Chile, with an emphasis on understanding the differentiation between populations on Chiloe Island versus mainland Chile. They sequence part of the mitochondrial control region and 12 microsatellite loci for nearly 200 individuals. They analyzed these datasets using various summary statistics and the programs Structure and GENELAND. A haplotype network of control region sequences shows relatively high genetic diversity, with some haplotypes being spread across wide geographic distances. I recommend supplementing this analysis with a phylogeny of control region sequences where geographic details are illustrated at the tips of the tree, which might demonstrate this pattern more effectively. The Structure analysis that assumes 5 populations shows a great deal of mixed ancestry for coastal temperate forest populations. Chiloe Island and Northern Patagonia populations are somewhat more consistent in population assignment. Overall, I agree with the authors that their Structure results do not show deep population structure. The GENELAND results confirm that there is no significant population structure between Chiloe and mainland populations.

Overall, I think this study relies on a solid dataset with good geographic and genetic sampling. It is modest in scope and, disappointingly, somewhat unfocused. The Introduction should be reframed around specific, testable hypotheses with clear predictions. It should also more clearly explain the geography of the region and the recent geological history. Some sections of the paper are clearly written (e.g., Methods and, for the most part, Results) but others (e.g. Discussion) seem incomplete or haphazardly reasoned.

For example, the focus on human impacts and conservation seems quite separate from the question of evolutionary drivers of population size changes and movements. The paragraph that starts at Line 340 discusses potential human impacts on population structure, and I found it difficult to parse. What, exactly, are the specific predictions we would expect to observe if human modification of the landscape affected the population structure of this species vs. more ancient drivers like glacial recession? Human impacts would necessarily be quite recent in evolutionary time, so it’s possible any impact would not be detected from control region or microsatellite data. This, of course, needs to be discussed.

The paper should be carefully proofread again and many sentences need to be revised because there were numerous small errors throughout. Typically, this did not prevent me from understanding the main results. Vague language, however, especially in the Discussion and Conclusion, did prevent me from taking away any clear message beyond the finding that there is little population structure between Chiloe and the mainland. One example of this vague, confusing language comes from Line 397 of the Conclusion, where the species is said to “reacts [sic] more to historical than to ecological events regarding its genetic population dynamics”. It’s not clear to me what evidence is used to justify this claim.

Other issues and suggestions are provided below:

Abstract

*Line 45: First sentence should be rephrased; not clear what “their impacts” is referring to.

Introduction

*Line 73: list is not consistently structured; “livestock” and “highways” are not activities
*Line 75: temperate rain forests, not just temperate forest, correct?
*Line 88: North population of what? No organism has been introduced yet. Should introduce the island of Chiloe here, too. What makes it distinctive or important?
*Line 98: “associated to” should be “associated with”; be sure to check throughout the text for that minor English language idioms like this are corrected.
*Line 101: how were these refuges recognized? And where are they?
*Line 104: misuse of “being” here and throughout. Rephrase.

Methods

*Line 196: which dataset was used for the Structure analysis? Just microsatellites?

Results

*Line 248: Authors state “there is an haplogroup (“light blue”) spread all over the network that corresponded to the locality of Valdivia”. I don’t know what this means. How are you defining “haplogroup”? By definition, a given haplotype can only be present in one place on a haplotype network.
*Line 279: Four clusters? This is explained confusingly. The Structure results suggest 5 clusters.
*Line 290: what is Micro-checker? Needs to be explained in the Methods

Discussion

*Line 299: The first sentence is not clear to me. Yes, it appears that there is high genetic diversity at this one locus across the species’ range. But how does the fact that there are few mutational steps between haplotypes translate into low nucleotide diversity? And then the next sentence starts with “thus” but I do not logically connect the first sentence to the second. Certainly, the haplotype network does not show clear division between Chiloe and the mainland—emphasize that point.
*Paragraph at Line 340: Discussion of potential human impacts on population structure is confusing. Human impacts would necessarily be quite recent in evolutionary time, so it’s possible any impact would not be detected from control region or microsatellite data. This needs to be discussed.

Figures

*Figure 1: light green lines are difficult to see. Draw map with black lines.

*Figure 2: this network is difficult to interpret on its own. Consider placing it on a map, or somehow denoting geography here.

---

## Round 0.2 · Minor Revisions

Although both reviewers think that your manuscript has been substantially improved, both provide additional suggestions that would need addressing in order to reach a final decision. I agree with Reviewer 3 in the need for a major revision of English writing. To ensure that the English language meets PeerJ standards, I suggest you ask a native English speaker to revise your text or use an academic copywriting service of your choosing.

In the methods (L225-231) you should mention which software was used to assess linkage disequilibrium between loci, and you should also provide the p-value for the Mantel test (L305) to assess its significance.

In Figure 1, you need to explain the meaning of dots with different colors.

In Table 1, the legend should explain why you are repeating columns and merging populations. Please explain which criteria was used to merge some populations, and only use the word “population” for genetic clusters. Sampling locations do not represent populations. I would also like to see an additional column with effective population size (Ne) for each genetic cluster, as this is an extremely relevant parameter cor conservation and management. You can use the NeEstimator software: http://www.molecularfisherieslaboratory.com.au/neestimator-software/

Table 2 should be a supplementary Table and I suggest to replace it with an isolation by distance figure: Fst/(1-Fst) ~ Geographic distance

I consider these minor yet important changes, and have henceforth asked for a minor revision.

Reviewer 2 ·

Basic reporting

I have read the revision version of this manuscript, and I found that the authors have addressed my comments adequately. I found the aims, analyses and results adequate, so I have no major comments. I only noticed some minor grammatical and small issues that should be addressed before its publication.

Ln56: clusters
Ln57: showed a genetic barrier in those areas, while the Chacao channel was not a significant barrier for this rodent.
Ln128: type of markers,…..may not be
Ln217: Add a comma after population structure
Ln230: Omit the comma after genetic differentiation
Ln232: Isolation by Distance (IBD) were calculated through Mantel test with GenAIEx
Ln234: Simple Euclidean algorithm?
Ln267: Delaunay
Ln305: test for IBD was positive and significant? .
Ln359: Microsatellite genetic diversity indices showed…
Ln388: an
Ln395: has been recognized
Ln407: Omit the comma after historical factors
Ln441-443: I would omit these sentences of results here, as this data was already mentioned. Just focus on giving general conclusions.

Experimental design

Adequate

Validity of the findings

Adequate

Reviewer 3 ·

Basic reporting

This manuscript has significant and persistent grammar and spelling issues. Almost every paragraph needs to be revised. Often, multiple sentences within a given paragraph contain typos, comma mistakes, or non-standard constructions. I noted this in my last review and see no attempt to fix these issues in the revision. I hope the authors are able to fix this because it is not publishable in its current form.

In general, the literature references are fine.

Figure 1 is much improved over the last version. The colors are helpful.

Experimental design

This is improved over the last version. I'm glad the authors eliminated the confusing focus on human impacts. The authors did not heed my recommendation from my previous review to include a phylogeny of control region sequences. I still think this would be helpful, even as a supplement. Phylogenies are routinely included in phylogeographic studies. It would be especially helpful to include sequences from related outgroup populations. This would put the subspecies in a phylogenetic context.

Validity of the findings

The dataset is solid and the conclusions are mostly reasonable. The paper is modest in scope.

Additional comments

This is my second review of this manuscript. The quality and clarity of the writing is the last major issue that must be resolved in my opinion. The addition of a phylogeny would also be helpful and very easy to do.

---

## Round 0.3 · Minor Revisions

Even though your revised manuscript has been improved I am not ready to accept if for publication untill my last requests are considered:

1) The legend of Figure 1 still does not explain the color code of sampling locations (green, light blue, dark blue, red, purple, gray and yellow dots). I understand you are referring to these colors in Figure 2, but each figure should be self-explanatory.

2) The legend of Table 1 still does not explain its content. Sampling locations should not be referred to as “populations”. There should be either two higher-level column labeled “Sampling location” and “Genetic cluster”, or you should remove genetic diversity statistics from sampling locations altogether, since these do not represent biologically meaningful units.

3) Table 2 should be placed in the supplementary material and the Mantel test plot brought to the main text ( Supplementary Figure 1 is actually Appendix 3, please correct).

Please also make sure all supplementary material is appropriately cited in the main text.

External reviews were received for this submission. These reviews were used by the Editor when they made their decision, and can be downloaded below.

---

## Round 0.4 · accepted · Accept

I am happy to accept the revised article since all the raised issues have been appropriately addressed.

# External reviews were received for this submission. These reviews were used by the Editor when they made their decision, and can be downloaded below.